# REVISITING PARAMETER SERVER IN LLM POST-TRAINING

**Xinyi Wan**[1,2][*], **Penghui Qi**[1,2][*], **Guangxing Huang**[1], **Chaoyi Ruan**[2], **Min Lin**[1] **& Jialin Li**[2]
[1]Sea AI Lab    [2]National University of Singapore

## ABSTRACT

Modern data parallel (DP) training favors collective communication over parameter servers (PS) for its simplicity and efficiency under balanced workloads. However, the balanced workload assumption no longer holds in large language model (LLM) post-training due to the high variance in sequence lengths. Under imbalanced workloads, collective communication creates synchronization barriers, leading to under-utilization of devices with smaller workloads. This change in training dynamics calls for a revisit of the PS paradigm for its robustness to such imbalance. We propose **On-Demand Communication (ODC)**, which adapts PS into Fully Sharded Data Parallel (FSDP) by replacing collective all-gather and reduce-scatter with direct point-to-point communication. Compared to FSDP, ODC reduces the synchronization barrier from once per layer to once per minibatch and decouples the workload on each device so that faster workers are not stalled. It also enables simpler and more effective load balancing at the minibatch level. Across diverse LLM post-training tasks, ODC consistently improves device utilization and training throughput, achieving up to a 36% speedup over standard FSDP. These results demonstrate that ODC is a superior fit for the prevalent imbalanced workloads in LLM post-training. Our implementation of ODC and integration with FSDP is open-sourced at `https://github.com/sail-sg/odc`.

## 1 INTRODUCTION

The development of DP distributed training (Krizhevsky, 2014; Goyal et al., 2017; Li et al., 2020) has followed two main approaches: the PS architecture and collective communication. Early large-scale systems such as DistBelief used the PS model to train deep neural networks across heterogeneous hardware and networks with variable latencies (Dean et al., 2012). In this setup, servers stored the model parameters while workers handled computation, enabling asynchronous or loosely synchronous training that tolerated slower or unreliable machines. Later work expanded on this design by enabling different consistency policies and exploring elastic scalability with continuous fault tolerance (Li et al., 2014). With the emergence of dense, homogeneous GPU clusters and high-bandwidth interconnects, collective communication became the mainstream approach for distributed DP. A prominent advantage of this paradigm was the opportunity it created for communication-efficient algorithms. Ring-based methods, as demonstrated in Baidu AllReduce (Research, 2017) and Horovod (Sergeev & Del Balso, 2018), reduced bandwidth requirements while scaling predictably. This trend was further reinforced by vendor-optimized libraries like NCCL (NVIDIA, b), which made high-performance collectives broadly accessible and easy to integrate into modern training frameworks. It is important to note that the high efficiency of collective communication fundamentally relies on balanced workloads. This presumption was largely valid for many dominant deep learning domains, including vision, speech, and early NLP. As a result, the dependency on workload balance was frequently taken for granted or neglected in system design.

Recently, the post-training of LLMs (Ouyang et al., 2022; Guo et al., 2025) breaks the long-standing assumption of balanced workloads that collective communication relies on. Real-world text corpora contain sequences of widely varying lengths (Bai et al., 2024; Yang et al., 2025). As the cost of attention grows quadratically with sequence length (Vaswani et al., 2017) while activation memory grows linearly, this variation leads to persistent computational imbalance across devices. Although

---

[*]Equal Contributors

a line of work has focused on mitigating this issue with sophisticated packing strategies (Krell et al., 2021; Kundu et al., 2024; Yao et al., 2025; Wang et al., 2025), these methods can only reduce the skew, but cannot remove it entirely, especially under memory constraints that force minibatches to be split into smaller microbatches (Huang et al., 2019; Qi et al., 2024). This not only narrows the solution space for effective packing, but also increases the number of synchronization points, further amplifying the inefficiency due to imbalanced workloads.

This inefficiency from workload imbalance is particularly severe in contemporary sharded DP, exemplified by ZeRO (Rajbhandari et al., 2020) and PyTorch's FSDP (Zhao et al., 2023). By sharding parameters, gradients, and optimizer states across devices, FSDP enables memory-efficient scaling to trillion-parameter models, making it the standard choice for LLM post-training and reinforcement learning (RL) pipelines (Hu et al., 2024; Sheng et al., 2025; Fu et al., 2025; Liu et al., 2024). However, this memory efficiency comes at the cost of increased synchronization (Figure 1). FSDP relies heavily on collective communication: per-layer parameters are reconstructed via *all-gather* before the forward pass, and gradients are aggregated via *reduce-scatter* after the backward pass. This fine-grained, layer-level synchronization implicitly assumes balanced workloads, which is precisely the assumption violated in LLM post-training. Our evaluation shows that even with state-of-the-art packing strategies, workload imbalance can still result in device idle times of up to 50% during long-sequence supervised fine-tuning (see Table 6).

To bridge the gap between fine-grained synchronization and workload imbalance in LLM post-training, we revisit the PS idea, and adapt it to the modern sharded DP paradigm through **On-demand Communication (ODC)**. We replace the per-layer collectives with point-to-point primitives, allowing devices to fetch parameters and push gradients independently (Figure 2). This reframes FSDP as a decentralized PS where server and worker roles are colocated, thus preserving its memory and scaling advantages. While preserving the synchronous optimization semantics, we relax synchronization from the layer level to the minibatch level. This decoupling of device progress significantly mitigates straggler effects and enables a more flexible space for workload balancing.

In summary, this paper presents a novel perspective: compared to collectives, the PS architecture is naturally better suited for LLM post-training due to its tolerance for heterogeneous workloads. To retain the key benefits of modern DP schemes, we do not build a standalone PS. Instead, we propose ODC, a communication scheme that brings the workload-tolerance of classic PS into FSDP. Our evaluation demonstrates that ODC substantially improves device utilization and end-to-end throughput across diverse LLM post-training tasks, including supervised fine-tuning (SFT) and RL, achieving up to 36% speedup over conventional FSDP.

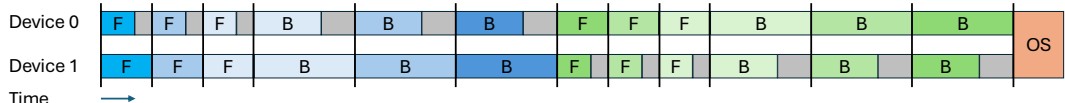

Figure 1: Collective communications introduces per-layer synchronization barriers in FSDP.

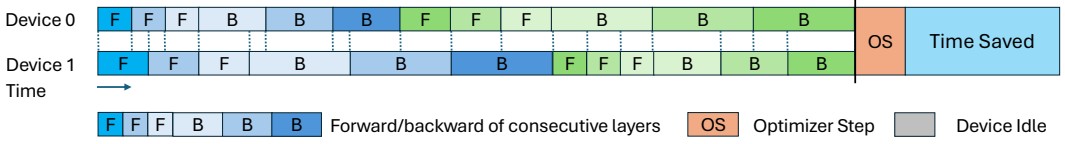

Figure 2: On-demand communications relaxes the synchronization barriers to minibatch end.

## 2 BACKGROUND

### 2.1 MINIBATCH, MICROBATCH AND GRADIENT ACCUMULATION

In deep learning, a minibatch refers to the set of training samples processed in a single optimizer step. However, training LLMs often exceeds the memory capacity required to process the desired

minibatch in one forward–backward pass. A common remedy is to divide the minibatch into $M$ microbatches and accumulate gradients before performing the optimizer update. For each microbatch $m \in 1, \ldots, M$, we compute the forward and backward passes to obtain per-parameter gradients $g^{(m)}$, and then accumulate $\bar{g} = \sum_{m=1}^{M} w_m g^{(m)}$, where $w_m$ encodes the aggregation policy (e.g., $w_m = 1$ for summation, or proportional weighting when averaging by tokens or samples).

## 2.2 SYNCHRONIZATION BARRIERS IN FSDP

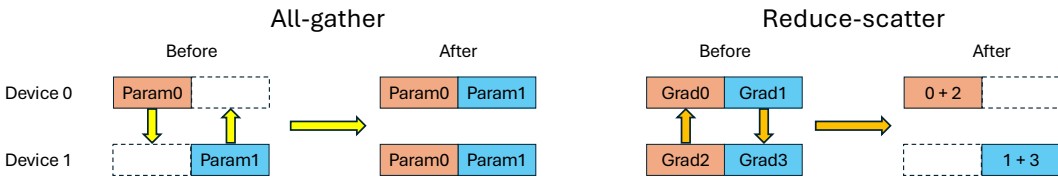

Figure 3: *all-gather* and *reduce-scatter*

In FSDP, both parameters and gradients are partitioned across devices. FSDP primarily uses *all-gather* to materialize parameters and *reduce-scatter* to aggregate gradients. The mechanics of *reduce-scatter* and *all-gather* are illustrated in Figure 3.

The communication pattern unfolds as follows. During the forward pass, before computation on a specific layer begins, its full parameters are reconstructed on each device via an *all-gather* operation. These reconstructed parameters are then discarded immediately after use to save memory. A similar *all-gather* process occurs during the backward pass. Additionally, after gradients are computed for a layer, they are aggregated and distributed using a *reduce-scatter* operation, leaving each device with only its corresponding shard of the total gradient. The overall communication flow is shown in Figure 4. In practice, modern implementations overlap these communications with computation (e.g., pre-fetching parameters for the next layer during the current layer's execution) to hide the latency, but this overlap does not remove the underlying synchronization points.

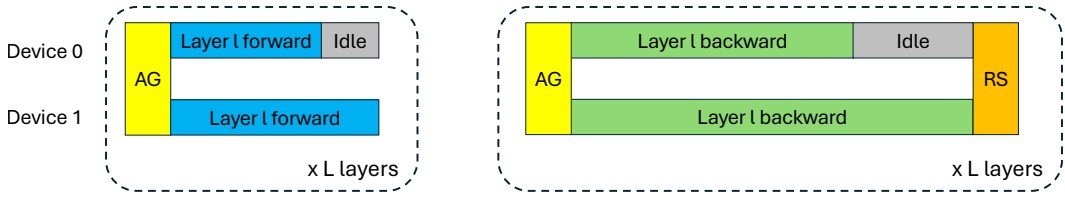

Figure 4: Communication pattern of FSDP within a microbatch. The left panel shows forward communication (*all-gather* parameters), and the right shows backward communication (*all-gather* parameters & *reduce-scatter* gradients). AG = *all-gather*; RS = *reduce-scatter*.

These per-layer collectives create fundamental synchronization barriers that are the root cause of inefficiency under imbalanced workloads. All devices must complete the *all-gather* before a layer's forward computation can begin, and they must all complete the *reduce-scatter* before gradient accumulation can proceed. This tight coupling forces all devices to advance at the same pace, meaning faster devices must idle and wait for the slowest one before moving to the next layer.

More formally, let a batching solution $\mathcal{P}_M$ specify the assignment of training samples to $M$ microbatches on each device. Denote by $T_{m,d,l}(\mathcal{P}_M)$ the time to execute layer $l$ of microbatch $m$ on device $d$ under $\mathcal{P}_M$. For a model with $L$ layers, the minibatch runtime is bounded by the slowest device at each per-layer step:

$$T(\mathcal{P}_M) = \sum_{m=1}^{M} \sum_{l=1}^{L} \max_d T_{m,d,l}(\mathcal{P}_M). \tag{1}$$

A significant body of research has focused on finding an optimal batching solution, $\mathcal{P}^\star$, that minimizes $T(\mathcal{P}_M)$. However, as we detail in Section 4, these approaches face fundamental limitations.

## 3 ON-DEMAND COMMUNICATIONS

To address the inefficiency of FSDP caused by imbalanced workload, we step back from the prevailing focus on complex batching strategies and re-examine a first principle of data parallelism: per-device computations are independent. Standard FSDP violates the spirit of this independence by using collective communication, which imposes fine-grained synchronization barriers. These barriers, which force devices to wait for the slowest one, are the direct cause of idle time. They are an artifact of the communication model, not a requirement of the training algorithm itself, and are therefore fundamentally **avoidable**.

To address this root cause, we propose ODC, a new communication scheme that relaxes synchronization to a much coarser granularity without altering the training semantics (Figure 2). ODC preserves FSDP's memory layout and computational graph but replaces its synchronous collectives with point-to-point operations. Specifically, we decompose the collective calls. An *all-gather* is replaced by a series of targeted *gather* requests, where a device fetches only the specific parameter shards it needs from its peers. Similarly, a *reduce-scatter* is broken down into a series of *scatter-accumulate* operations, where a device pushes its computed gradients directly to the devices that own the corresponding gradient shards. This process is illustrated in Figures 5. With ODC, each device operates independently, fetching parameters or pushing gradients as soon as it is ready, thereby eliminating the synchronization-induced stalls.

A critical feature of ODC is that these point-to-point data transfers are non-intrusive. When one device initiates a *gather* or *scatter-accumulate* request to another, it does not interrupt the ongoing computation on the target device. We show how this is enabled in Section 3.2.

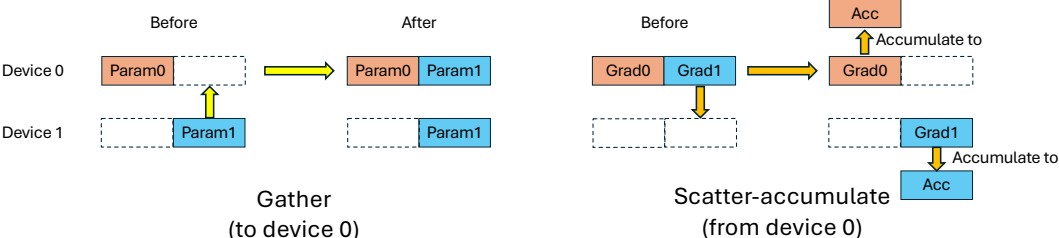

Figure 5: *gather* and *scatter-accumulate*.

### 3.1 ODC AS A DECENTRALIZED PARAMETER SERVER

The classic PS architecture (Dean et al., 2012; Li et al., 2014) separates model state from model computation, where a set of server nodes is responsible for storing the model's parameters and optimizer states. Meanwhile, a set of worker nodes pulls parameters from the servers, performs the forward and backward computations on its local data, and then pushes the resulting gradients back to the servers. The servers then aggregate these gradients and apply the updates. This design decouples the progress of individual workers and provides a natural tolerance for stragglers, which is a key advantage for the imbalanced workloads common in LLM post-training.

As shown in Figure 6, ODC paradigm reframes FSDP as a modern, decentralized PS. Instead of using dedicated server nodes, we colocate the server and worker roles by evenly partitioning parameters, gradients, and optimizer states across all devices. Each device acts as a server by owning and managing a shard of the model's parameters and optimizer state. Simultaneously, it acts as a worker by executing the forward and backward passes on its assigned data. This decentralized, co-located design mirrors the memory layout of FSDP and avoids the network bottlenecks of a centralized PS. While colocated roles has precedent in some PS systems (Jiang et al., 2020), our approach is novel in its direct integration with FSDP's sharding mechanism.

Ultimately, by replacing FSDP's per-layer collectives with on-demand point-to-point communication, our method gains the imbalance tolerance of a PS while retaining the core benefits of FSDP: memory efficiency, decentralization, scalability, and simplicity.

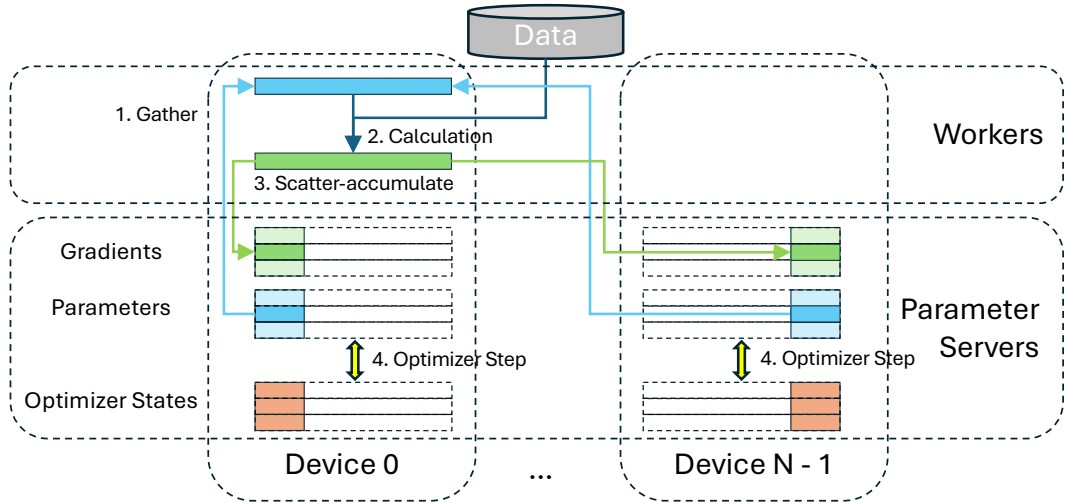

Figure 6: The architecture of FSDP with ODC, in which FSDP can be seen as a decentralized parameter server, with server part and worker part highlighted in this figure.

## 3.2 IMPLEMENTATION

ODC workers often push or pull data to servers while colocated workers concurrently perform computations, making it essential to minimize server interference. Communication primitives must also support ODC's on-demand nature, where workers control the flow and servers cannot anticipate requests. Existing message-based libraries like MPI (Gabriel et al., 2004) and NCCL(NVIDIA, b) require explicit, ordered participation from both sender and receiver, making them neither transparent nor on-demand, and prone to deadlocks if not carefully scheduled.

ODC instead leverages native RDMA-based interfaces: CUDA IPC (NVIDIA, a) for intra-node and NVSHMEM (NVIDIA, c) for inter-node communication. RDMA enables transparent data transfers without active server involvement, except for gradient accumulation, which is handled by a lightweight daemon. The communication kernel is built on Triton-Distributed (Zheng et al., 2025), a Triton (Tillet et al., 2019) wrapper that exposes RDMA functionalities directly in Python Triton kernels, eliminating the need for low-level CUDA C code. We put more implementation details at Appendix B, and will open-source our implementation for community usage.

Integrating ODC into FSDP is straightforward: it only requires replacing collective communication calls with ODC primitives and retrieving accumulated gradients at the minibatch end.

## 4 SIMPLIFIED LOAD BALANCING WITH ODC

Due to the variation in sequence lengths, a naive padding strategy significantly suffers from computation waste. To mitigate this, Krell et al. (2021) introduced the strategy of sequence packing, which concatenates multiple samples into a single sequence with appropriate attention masks, improving utilization and balancing workload across microbatches. This approach has been broadly adopted and extended by subsequent work (Bai et al., 2024; Kundu et al., 2024; Yao et al., 2025; Wang et al., 2025), with efficient support in modern libraries like FlashAttention (Dao et al., 2022; Dao, 2023).

However, existing sequence packing methods operate at the microbatch level, which faces several fundamental limitations under FSDP. First, the size of a microbatch is bounded by device memory, limiting the number of samples per microbatch and leaving substantial variance in workload across devices. This effect is amplified in long-sequence training regimes, such as LongAlign (Bai et al., 2024) and RL for LLM reasoning (Guo et al., 2025), where extended contexts further constrain per-device capacity. Second, for a sample of sequence length $s$, activation memory typically scales as $O(s)$ while runtime scales as $O(s^2)$ (e.g., due to attention), creating a fundamental mismatch between memory and compute. Consequently, compute alignment can be infeasible under memory

constraints. For instance, if a microbatch contains a single sample at the maximum sequence length, no feasible packing of shorter samples can match its runtime.

By replacing collective operations with ODC, our approach decouples the execution of microbatches across devices. This eliminates synchronization barriers inherent in FSDP and removes the implicit requirement for a uniform number of microbatches per device. This insight allows for a significant simplification of workload balancing strategy. Specifically, our strategy shifts the balancing objective from the fine-grained microbatch level to the coarser minibatch level. We first partition the global set of training samples across devices with the sole goal of balancing the total computational load. Subsequently, each device independently packs its local subset of samples into microbatches, governed only by its local memory constraints. This shift in granularity not only simplifies the packing algorithm, but also achieves superior load balancing by operating on a larger, less constrained set of samples. We leave the detailed packing algorithms in Appendix C.

## 5 EVALUATIONS

### 5.1 SETUP

We evaluate ODC on two major LLM post-training tasks: SFT and RL. For SFT, we use a) LongAlign (Bai et al., 2024), a dataset for extending LLM context windows, and b) open-source trajectories from SWE-Smith (Yang et al., 2025), an agent model for software engineering tasks released by the SWE-Bench team (Jimenez et al., 2023). For RL, we run GRPO (Guo et al., 2025; Liu et al., 2025) implemented in verl (Sheng et al., 2025) on AIME prompts (Li et al., 2024), which includes problems from Olympiad-level math contest. Notably, we only record the model training time in RL, ignoring forward-only parts like actor rollout. The sequence length distributions of these datasets are shown in Figure 7.

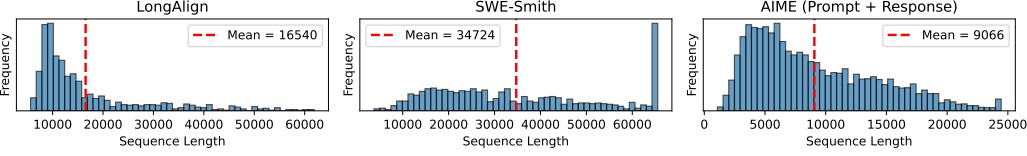

Figure 7: Sequence length distributions of evaluation datasets.

We evaluate ODC on the DeepSeek-R1-Distill-Qwen family of models (Team, 2024; Guo et al., 2025), with varying size from 1.5B to 32B. The models are trained on up to 32 NVIDIA A100 80G GPUs, with NVSwitch for intra-node communication and RoCE RDMA (800 Gbps per node) for inter-node communication. Notably, for RL experiment we run only up to 14B model using 16 GPUs, as the inference time would be too long for a 32B model. Additionally, we validate the correctness of ODC by verifying the training convergency in Appendix F.

Each method in our evaluation is a combination of communication scheme and load balancing algorithms. For communication scheme, we have a) *Collective* - baseline using collective *all-gather* and *reduce-scatter*; b) *ODC* - our approach introduced in Section 3; For load balance algorithms, we include a) *LocalSort* - adapted from Bai et al. (2024); within each device's minibatch, sequences are sorted by length but not packed. b) *LB-Micro* - a heuristic-based packing baseline designed to minimize workload imbalance across devices within the same microbatch. In RL experiments, we show that it is substantially faster than the native implementation in verl (Sheng et al., 2025), underscoring its effectiveness as a strong baseline. c) *LB-Mini* - our algorithm introduced in Section 4, which balances workload at the minibatch level. As LB-Mini can produce different number of microbatches for different devices, it applies only to ODC. Detailed implementations can be found in Appendix C. Unless otherwise specified, the maximum number of tokens in a microbatch is constrained by the maximum sequence length of a single sample in the dataset.

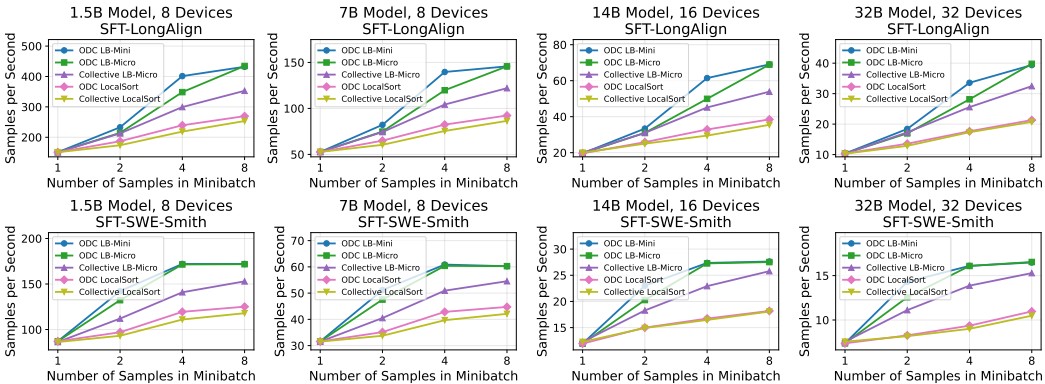

Figure 8: Samples per second on SFT datasets (LongAlign and SWE-Smith) across different model scales and minibatch sizes. ODC consistently improves throughput over Collectives in both unpacked (LocalSort) and packed (LB-Micro, LB-Mini) scenarios.

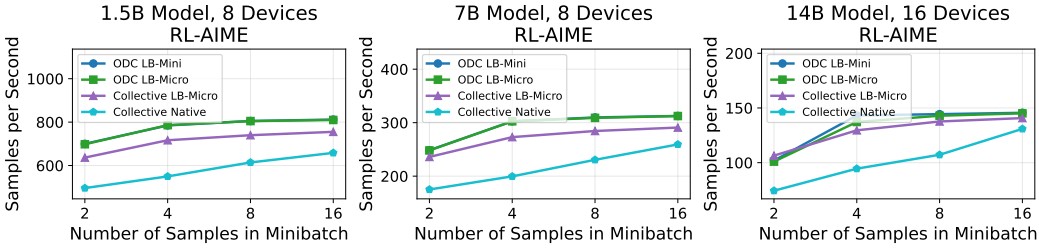

Figure 9: Samples per second on RL with AIME prompts. In addition to the methods in Section 5.1, we also evaluate the default load balancing algorithm in verl, denoted as *Native*. LB-Micro is substantially faster than Native, underscoring its effectiveness as a strong baseline.

## 5.2 MAIN RESULTS

Figure 8 presents the evaluation results on SFT tasks. ODC consistently improves throughput over the collective baseline in both unpacked (LocalSort) and packed (LB-Micro, LB-Mini) settings, with the most pronounced gains observed under packing, reaching up to a 36% speedup. All methods perform similarly when the minibatch size is one, since in this case ODC synchronizes after every sample, just like collective.

Figure 9 shows in RL tasks ODC achieves up to 10% speedup over collective baseline, although the gains are less pronounced than in SFT. This is primarily due to: a) implementation constraints in verl, which require identical numbers of samples per device and thus limit the effectiveness of LB-Mini. While relaxing this constraint is feasible, we did not do so, as the current solution is easier to integrate; and b) a less long-tailed sequence length distribution compared to SFT datasets (Figure 7).

At small minibatch sizes, LB-Mini often outperforms LB-Micro. This reflects the benefits of its minibatch-level balancing, which permits devices to process different numbers of microbatches. As the minibatch size increases, however, LB-Micro has more flexibility to balance workloads effectively, which narrows the performance gap between the two methods. The detailed timing data as well as bubble rate is reported in Appendix G.

## 5.3 PARAMETRIC STUDY

The effectiveness of ODC compared to collectives depends on several factors: a) Minibatch size: the number of samples per minibatch per device; b) Max length: the maximum sequence length in the dataset; to control this factor while maintaining the overall distribution, we adjust each sample by uniformly truncating or repeating tokens at a fixed ratio; c) Packing ratio: the maximum number

of tokens allowed in a microbatch divided by the max sequence length (e.g., with a max sequence length of 16K and packing ratio of 2, a microbatch may contain up to 32K tokens); d) Devices: the total number of devices.

To isolate the impact of each factor, we adopt a controlled methodology: starting from a fixed golden setting (Table 1), we vary one factor at a time while holding others constant. As shown in Figure 10, the acceleration ratio peaks at moderate minibatch sizes before declining as larger batches give the baseline more flexibility; it increases with sequence length, since longer sequences amplify the quadratic compute cost and exacerbate imbalance; it decreases with packing ratio, which improves the baseline's packing efficiency; and it grows with the number of devices, as more devices introduce greater heterogeneity.

| Model | Dataset | minibatch Size | Devices | Packing Ratio |
|-------|---------|----------------|---------|---------------|
| 1.5B | LongAlign (Max 64K) | 4 | 8 | 1 |

Table 1: Golden setting for the parametric study. Each experiment varies at most one factor.

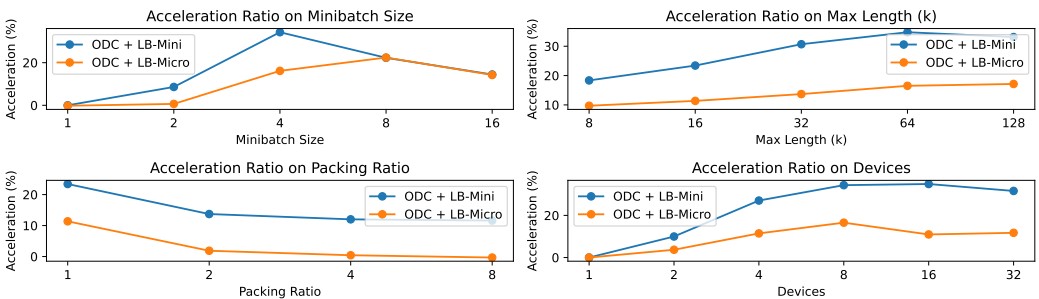

Figure 10: Acceleration ratio of ODC compared to collective with LB-Micro in parametric study.

## 5.4 BENCHMARK ON COMMUNICATION PRIMITIVES

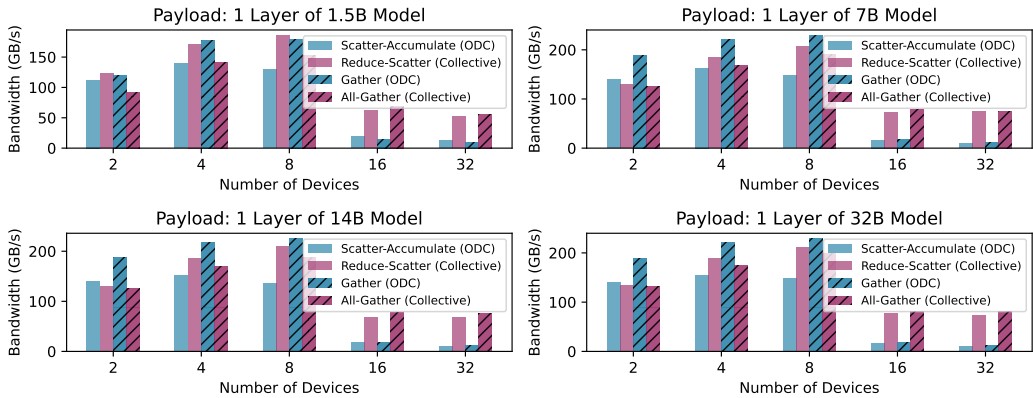

Figure 11: Benchmarking communication primitives against collectives. Within a node, ODC has a comparable performance with collective. But significantly slower than collective cross node.

We compare the bandwidth of ODC primitives (*gather* and *scatter-accumulate*) against collectives (*all-gather* and *reduce-scatter*) in NCCL. For fairness, ODC primitives are launched synchronously: each device issues operations in the same order, with barriers inserted before and after each primitive. Results are shown in Figure 11. Within a single node (up to 8 devices), ODC achieves bandwidth comparable to collective. However, once communication spans multiple nodes, ODC lags significantly behind collective. We leave more discussion and how to mitigate this inter-node inefficiency in Section 6.

## 6 DISCUSSION

### 6.1 CHALLENGES ON INTER-NODE COMMUNICATION EFFICIENCY

Collective primitives are often highly optimized by exploiting hierarchical interconnects in multi-node settings. For example, an all-gather operation might first perform an inter-node broadcast followed by an intra-node broadcast to minimize costly inter-node traffic. ODC does not increase communication volume, but changes the topology: it uses point-to-point RDMA and thus forgoes these hierarchical optimizations (see Appendix D). However, we argue that larger DP scale typically amplifies straggler effects under imbalance, increasing the benefit of ODC's decoupled progress (see Figure 10). Furthermore, several ways can effectively mitigate this communicate overhead.

**Overlapping Communication with Computation.** ODC retains the standard FSDP optimization of overlapping communication with computation. This is particularly effective because communication volume per microbatch is constant with sequence length ($s$), whereas computation scales as $O(s^2)$. For long sequences, the large computational cost effectively hides the communication latency. Consequently, despite using a non-hierarchical communication pattern, ODC shows no significant slowdown in our long-context evaluations (see Section 5.2).

**Hybrid Sharding.** When the tokens per microbatch is too small to hide communication costs, hybrid sharding provides an effective solution. Similar to ZeRO++ (Wang et al., 2024), parameters and gradients are sharded only *within* a node, while optimizer states remain sharded *across* nodes. This design eliminates cross-node parameter *gather* and gradient *scatter-accumulate*, at the cost of higher per-node memory usage, which is a manageable trade-off given that activation memory requirements are lower. As shown in Appendix E, this strategy effectively mitigates ODC's additional overhead.

### 6.2 FUTURE WORK

ODC is an initial effort toward adapting PS to modern sharded DP. We believe this is a foundational step that opens several promising directions for future research.

**ODC-specific Optimizations** While our current ODC implementation uses direct point-to-point communication, its communication graph can be further optimized. For instance, a device could fetch a parameter shard from a peer on the same node that has already cached it, effectively creating a hierarchical communication path similar to topology-aware collectives.

**Relaxing Synchronization Guarantees** Our current design intentionally preserves a synchronous update at the minibatch boundary to maintain identical training semantics. However, this barrier could be relaxed. Extending ODC to support classic asynchronous SGD schemes (Recht et al., 2011), such as bounded-staleness updates (Chen et al., 2016; Ho et al., 2013), could further reduce idle time and improve hardware utilization, particularly in highly heterogeneous environments. This would, however, require a careful analysis of the convergence implications for LLM training.

**Elasticity and Fault Tolerance** A significant advantage of PS-style architectures is their natural support for elasticity and fault tolerance (Dean et al., 2012; Li et al., 2014). Collective-based systems, in contrast, are notoriously brittle and difficult to resize (Jiang et al., 2020; Narayanan et al., 2021; Duan et al., 2024). Integrating these capabilities into ODC would improve the resilience and flexibility of large-scale, long-running LLM training jobs.

## 7 CONCLUSION

This paper revisits PS and adapts its principles to solve a critical bottleneck in modern sharded DP training for LLM post-training. We identified that the per-layer *all-gather* and *reduce-scatter* collectives in FSDP create fine-grained synchronization barriers, which amplify the straggler effects caused by workload imbalance.

We proposed ODC to replace these collectives with point-to-point operations, effectively relaxing synchronization from the layer level to the minibatch level. This approach, which reframes FSDP as a decentralized PS, decouples device execution and enables more effective load balancing. Empirically, ODC delivers consistent throughput and utilization gains across a range of long-sequence SFT and RL tasks.

## REPRODUCIBILITY STATEMENT

To ensure reproducibility of our experiments, we open-source our implementation, including: a) the core communication library of ODC, and b) the code patch that integrates ODC into FSDP at `https://github.com/sail-sg/odc.`.

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

## A LLM USAGE STATEMENT

LLMs were used to polish the writing of this paper and to assist in generating code for producing graphs used to present the evaluation results.

## B IMPLEMENTATION DETAILS OF ODC

For intra-node communication, we use CUDA-IPC (NVIDIA, a), which supports native read/write operations on remote GPU tensors as if they were local. As a result, no custom GPU kernels are required for intra-node communication. For inter-node communication, we implement a custom kernel using the `put_mem` and `get_mem` primitives provided by Triton-Distributed (Zheng et al., 2025).

The implementation of *gather* is straightforward: each rank pulls data from all other ranks using `get_mem`. Empirically, we find that limiting the communication payload per transfer helps stabilize RDMA traffic, as a server may receive RDMA read requests from multiple clients simultaneously.

The implementation of *scatter-accumulate* is slightly more involved. After a worker pushes data to a server using `put_mem`, it notifies the server over the same RDMA channel. The server runs a lightweight daemon process that polls for notifications and performs gradient accumulation upon receipt. As the polling does not occupy GPU SMs, we see no observable slowdown of the colocated worker (compute) process. Because a server can receive concurrent pushes from multiple clients, we allocate a dedicated buffer for each client to enable parallel data transfers and maximize throughput. Since requests from any single client are serialized, only one buffer per client is required. This design bounds the buffer memory on each server to $M/N$ per client, resulting in a total of $M/N \times N = M$ per server, where $N$ is the number of GPUs and $M$ is the number of elements in a transformer layer.

## C SEQUENCE PACKING STRATEGIES USED IN EXPERIMENT

We use the Karmarkar-Karp algorithm (Karmarkar & Karp, 1982) to balance computational workloads by solving the number partitioning problem[1]. Our approach builds on the implementation in the Verl framework (Sheng et al., 2025) but adds a crucial modification to prevent out-of-memory (OOM) errors. We modify the *microbatch_partition* function to iteratively validate that any proposed partition is memory-feasible before accepting it as a solution. This ensures robust execution even in memory-constrained settings. The key implementation details are provided in Listing 1.

### C.1 LB-MICRO AND LB-MINI

As detailed in Section 5.1, we compare two primary load-balancing strategies: LB-Micro and LB-Mini. LB-Micro performs workload balancing at the microbatch level, adhering to the conventional constraint that all devices must process an identical number of microbatches. In contrast, LB-Mini, enabled by our ODC framework, balances the workload at the coarser minibatch level. The fundamental advantage of LB-Mini is that it removes the rigid constraint on the number of microbatches per device, allowing for more flexible and effective load distribution. We highlight the implementation differences between these two approaches in Listing 1.

### C.2 VERL NATIVE TWO-LEVEL PARTITIONING STRATEGY

The packing method in the Verl framework is subject to two main constraints. First, it assumes that the number of training samples assigned to each device is the same. Second, because of layer-level synchronization, all devices must process an equal number of microbatches. For these reasons, Verl uses a two-level hierarchical heuristic approach, the implementation of which can be found in Listing 2.

---

[1] https://en.wikipedia.org/wiki/Partition_problem

### C.3 OPTIMIZED TWO-LEVEL PARTITIONING STRATEGY

The native partitioning strategy in Verl is suboptimal because it balances workloads at the global batch level, prior to splitting the data into minibatches. This approach fails to ensure balance within each individual minibatch. To correct this, we optimize the implementation by first partitioning the data into minibatches and then performing load balancing across devices for each minibatch. This reversal yields substantial throughput improvements, as shown in Figure 9. Our optimized implementation is detailed in Listing 3.

Listing 1: Helper functions.

```
def karmarkar_karp(
    compute_costs: List[int],    # Input list of computational costs
    k_partitions: int,           # The target number of partitions
    equal_size: bool             # If true, enforce equal size
) -> List[List[int]]:            # Returns the partitions of indexes
    """Split input into partitions to balance workload"""

def get_compute_costs(seqlen_lst: List[int]) -> List[int]:
    """Get the compute costs given the sequence lengths."""

def check_oom(micro_seqlen_lst: List[int]) -> int:
    """Check if the microbatch will OOM; returns 1 if OOM, else 0."""

def minibatch_partition(
    global_seqlen_lst: List[int], world_size: int
) -> List[List[int]]:
    compute_costs = get_compute_costs(global_seqlen_lst)
-   equal_size = True
+   equal_size = False # set False for SFT with ODC+LB_Mini
    partition_lst = karmarkar_karp(
        compute_costs, k_partitions=world_size, equal_size=equal_size)
    return partition_lst

def microbatch_partition(
    minibatch_seqlen_lst: List[int]
) -> List[List[int]]:
    minibatch_compute_costs = get_compute_costs(minibatch_seqlen_lst)
    k_partitions = 1
    while True:
        microbatch_partition_lst = karmarkar_karp(
            minibatch_compute_costs, k_partitions, equal_size=False)
        is_oom = check_oom(minibatch_seqlen_lst)
-       same_micro_in_dp = True
+       same_micro_in_dp = False # set False for ODC+LB_Mini
        if same_micro_in_dp:
            # Ensure all ranks have equal number of microbatches.
            torch.distributed.all_reduce(is_oom)
        if is_oom == 0:
            break # Found a valid packing configuration.
        k_partitions += 1
    return microbatch_partition_lst
```

Listing 2: Pseudocode for workload balancing and packing algorithms.

```
def verl_native_ppo_step(
    global_seqlen_lst: List[int], world_size: int, minibatch_size: int
):
    """PPO step with two-level partitioning."""
    global_seqlen_np = np.array(global_seqlen_lst)
    # Step 1: Balance global batch across ranks.
    rank_to_ppobatch = minibatch_partition(global_seqlen_lst, world_size)
    # The following block runs in parallel on each device.
    for rank, ppobatch in enumerate(rank_to_ppobatch):
```

```
        # run in parallel
        # For simplicity, we assume PPO epoch is 1.
        ppobatch = shuffle(ppobatch)
        # Step 2: Split local batch into minibatches.
        minibatches = [ppobatch[i:i + minibatch_size]
            for i in range(0, len(ppobatch), minibatch_size)]
        for minibatch in minibatches:
            minibatch_seqlen_lst = global_seqlen_np[minibatch].tolist()
            # Step 3: Partition minibatch into microbatches.
            microbatch_partition_lst = microbatch_partition(
                minibatch_seqlen_lst)
            for microbatch in microbatch_partition_lst:
                # Perform PPO update on the microbatch.
                ...
```

Listing 3: Pseudocode for workload balancing and packing algorithms.

```
def verl_optimized_ppo_step(
    global_seqlen_lst: List[int], world_size: int, minibatch_size: int
):
    """PPO step with optimized two-level partitioning."""
    # For simplicity, we assume PPO epoch is 1.
    global_seqlen_lst = shuffle(global_seqlen_lst)
    minibatch_size *= world_size
    # Step 1: Split global data into minibatches
    global_minibatches = [
        global_seqlen_lst[i:i + minibatch_size]
        for i in range(0, len(global_seqlen_lst), minibatch_size)
    ]
    for global_minibatch in global_minibatches:
        global_seqlen_np = np.array(global_minibatch)
        # Step 2: Balance minibatch across ranks.
        rank_to_minibatch = minibatch_partition(
            global_minibatch, world_size)
        for rank, minibatch in enumerate(rank_to_minibatch):
            # run in parallel
            minibatch_seqlen_lst = global_seqlen_np[minibatch].tolist()
            # Step 3: Partition minibatch into microbatches.
            microbatch_partition_lst = microbatch_partition(
                minibatch_seqlen_lst)
            for microbatch in microbatch_partition_lst:
                # do ppo update
                ...
```

## D  COMMUNICATION VOLUME COMPARISON

| Method | Intra-node | Inter-node | Total |
|---|---|---|---|
| Collective *all-gather* (Ring) | $\frac{G-1}{G} * (D-1) * K$ | $\frac{1}{G} * (D-1) * K$ | $(D-1) * K$ |
| ODC *gather* | $(G-1) * K$ | $(D-G) * K$ | $(D-1) * K$ |
| Collective *reduce-scatter* (Ring) | $\frac{G-1}{G} * (D-1) * K$ | $\frac{1}{G} * (D-1) * K$ | $(D-1) * K$ |
| ODC *scatter-accumulate* | $(G-1) * K$ | $(D-G) * K$ | $(D-1) * K$ |

Table 2: Comparison of per-client communication volume for collectives and ODC. Both methods send the same total volume ($(D-1) * K$), but ODC increases inter-node traffic since clients *gather/scatter-accumulate* independently.

Let $D$ denote the total number of devices, $G$ the number of devices per node, and $K$ the size of the per-device local parameters or gradients. Under these assumptions, Table 2 summarizes the per-client communication volume for collectives versus ODC. We can observe that ODC increases the cross-node communication volume, which might in turn slows down end-to-end communication.

# E  ZERO++ STYLE HYBRID SHARDING

We evaluate the hybrid sharding strategy introduced in Section 6.1, which is particularly effective for shorter sequence lengths. To simulate this setting, we truncate each sequence in LongAlign to $1/8$ of its original length, resulting in a dataset with a maximum length of 8k and an average length of 2k. As shown in Figure 12, hybrid sharding achieves acceleration comparable to full sharding—up to 28%—when comparing ODC against collectives.

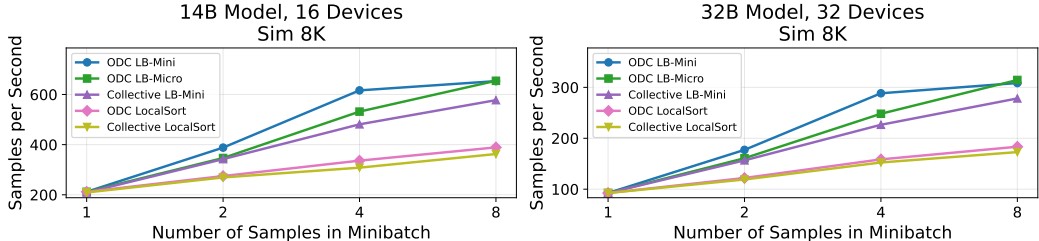

Figure 12: Comparing ODC with Collectives using hybrid sharding.

It is worth noting that hybrid sharding incurs higher memory usage compared to fully sharded training; a detailed memory usage comparison is provided in Figure 13.

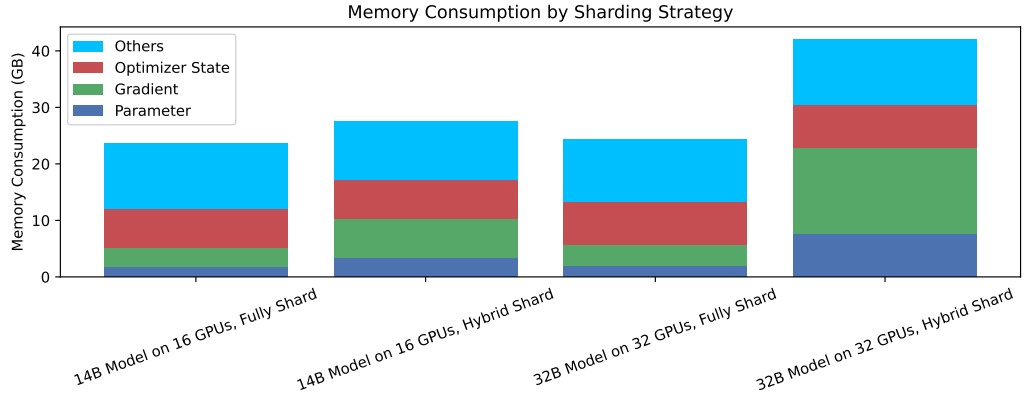

Figure 13: Memory consumption of ODC in hybrid and full sharding.

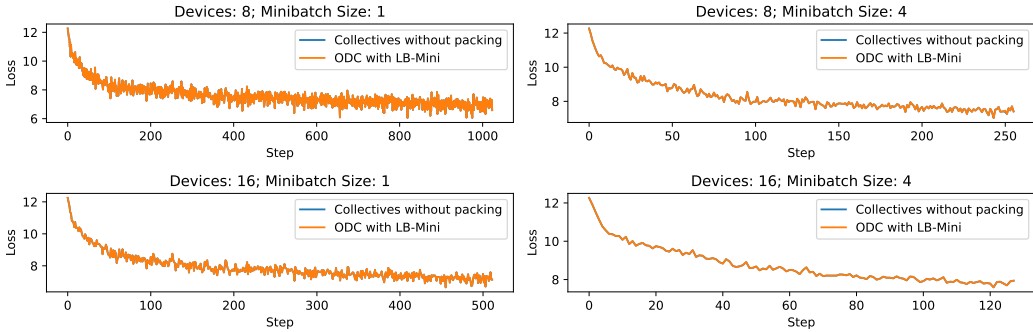

Figure 14: Training loss curves on 8k samples from LongAlign with a 1.5B model. ODC and Collective produce almost identical loss curves.

## F  CONVERGENCY VERIFICATION

To validate the correctness of our implementation, we compare the loss curves when training a 1.5B model from scratch (to produce a clearer loss-descent trajectory) on 8k samples from LongAlign. Results are shown in Figure 14.

## G  DETAILED EXPERIMENT DATA

We show the detailed timing data for SFT and RL in Table As shown in Tables 5 and 3, we also report the bubble rate, defined as the ratio of device idle time—caused by workload imbalance—to the total run time, as estimated by the packing algorithm (Tables 6 and 4).

We observe that the acceleration achieved by ODC closely correlates with the bubble rate predicted by the packing algorithms, indicating that the performance gains primarily stem from reducing idle time due to workload imbalance.

| Model | Dataset | Method | Samples Per Second Per Device | | | |
|---|---|---|---|---|---|---|
| | | | Minibs=2 | 4 | 8 | 16 |
| 1.5B | AIME | Collective Native | 496.1 | 549.9 | 614.1 | 658.3 |
| 1.5B | AIME | Collective LB-Micro | 636.6 | 716.4 | 739.7 | 755.2 |
| 1.5B | AIME | ODC LB-Micro | 698.0 (+10%) | 786.0 (+10%) | 804.8 (+9%) | 809.9 (+7%) |
| 1.5B | AIME | ODC LB-Mini | 700.0 (+10%) | 784.6 (+10%) | 805.1 (+9%) | 811.8 (+7%) |
| 7B | AIME | Collective Native | 175.1 | 199.6 | 230.5 | 259.3 |
| 7B | AIME | Collective LB-Micro | 235.9 | 273.0 | 284.4 | 290.8 |
| 7B | AIME | ODC LB-Micro | 248.1 (+5%) | 302.6 (+11%) | 309.0 (+9%) | 312.4 (+7%) |
| 7B | AIME | ODC LB-Mini | 248.5 (+5%) | 302.5 (+11%) | 309.8 (+9%) | 312.6 (+7%) |
| 14B | AIME | Collective Native | 74.4 | 94.5 | 107.2 | 130.9 |
| 14B | AIME | Collective LB-Micro | 106.4 | 129.6 | 137.6 | 140.7 |
| 14B | AIME | ODC LB-Micro | 101.0 (-5%) | 137.2 (+6%) | 142.9 (+4%) | 145.1 (+3%) |
| 14B | AIME | ODC LB-Mini | 101.9 (-4%) | 143.1 (+10%) | 144.4 (+5%) | 145.6 (+3%) |

Table 3: Timing Data for RL. For the percentage, ODC LB-Micro and ODC LB-Mini is comparing against Collective LB-Micro, ODC LocalSort is comparing against Collective LocalSort.

| Model | Devices | Dataset | Method | Bubble Rate (%) | | | |
|---|---|---|---|---|---|---|---|
| | | | | Minibs=2 | 4 | 8 | 16 |
| 1.5B | 8 | AIME | Collective LB-Micro | 15.73 | 6.56 | 3.47 | 1.65 |
| 1.5B | 8 | AIME | Collective Native | 33.83 | 27.61 | 20.81 | 13.13 |
| 1.5B | 8 | AIME | ODC LB-Micro | 10.26 | 0.51 | 0.05 | 0.01 |
| 1.5B | 8 | AIME | ODC LB-Mini | 10.26 | 0.51 | 0.05 | 0.01 |
| 7B | 8 | AIME | Collective LB-Micro | 20.79 | 7.48 | 3.93 | 1.90 |
| 7B | 8 | AIME | Collective Native | 40.15 | 32.41 | 23.40 | 13.45 |
| 7B | 8 | AIME | ODC LB-Micro | 16.85 | 0.53 | 0.06 | 0.01 |
| 7B | 8 | AIME | ODC LB-Mini | 16.85 | 0.53 | 0.06 | 0.01 |
| 14B | 16 | AIME | Collective LB-Micro | 28.35 | 10.77 | 5.91 | 2.48 |
| 14B | 16 | AIME | Collective Native | 46.68 | 36.36 | 26.63 | 14.83 |
| 14B | 16 | AIME | ODC LB-Micro | 22.89 | 0.61 | 0.04 | 0.01 |
| 14B | 16 | AIME | ODC LB-Mini | 22.89 | 0.61 | 0.04 | 0.01 |

Table 4: Bubble Rate Data for RL

| Model | Dataset | Method | Samples Per Second Per Device | | | |
|-------|---------|--------|----------|---|---|---|
| | | | Minibs=1 | 2 | 4 | 8 |
| 1.5B | LongAlign | Collective LocalSort | 150.7 | 173.8 | 218.7 | 253.4 |
| 1.5B | LongAlign | ODC LocalSort | 150.9 (+0%) | 186.9 (+8%) | 239.5 (+10%) | 269.5 (+6%) |
| 1.5B | LongAlign | Collective LB-Micro | 150.7 | 212.8 | 299.4 | 352.9 |
| 1.5B | LongAlign | ODC LB-Micro | 150.9 (+0%) | 214.4 (+1%) | 348.5 (+16%) | 434.6 (+23%) |
| 1.5B | LongAlign | ODC LB-Mini | 150.9 (+0%) | 232.9 (+9%) | 401.3 (+34%) | 432.0 (+22%) |
| 1.5B | SWE-Smith | Collective LocalSort | 86.3 | 93.1 | 111.0 | 117.9 |
| 1.5B | SWE-Smith | ODC LocalSort | 87.1 (+1%) | 97.0 (+4%) | 119.4 (+8%) | 125.1 (+6%) |
| 1.5B | SWE-Smith | Collective LB-Micro | 86.3 | 112.0 | 140.9 | 152.9 |
| 1.5B | SWE-Smith | ODC LB-Micro | 87.1 (+1%) | 132.2 (+18%) | 171.5 (+22%) | 171.7 (+12%) |
| 1.5B | SWE-Smith | ODC LB-Mini | 87.1 (+1%) | 142.2 (+27%) | 172.0 (+22%) | 172.0 (+12%) |
| 7B | LongAlign | Collective LocalSort | 52.7 | 60.4 | 75.5 | 86.4 |
| 7B | LongAlign | ODC LocalSort | 52.6 (-0%) | 64.9 (+8%) | 82.4 (+9%) | 92.3 (+7%) |
| 7B | LongAlign | Collective LB-Micro | 52.7 | 74.3 | 104.2 | 122.0 |
| 7B | LongAlign | ODC LB-Micro | 52.6 (-0%) | 74.8 (+1%) | 119.8 (+15%) | 145.7 (+19%) |
| 7B | LongAlign | ODC LB-Mini | 52.6 (-0%) | 82.1 (+11%) | 139.6 (+34%) | 145.7 (+19%) |
| 7B | SWE-Smith | Collective LocalSort | 31.5 | 33.7 | 39.7 | 42.1 |
| 7B | SWE-Smith | ODC LocalSort | 31.6 (+0%) | 35.1 (+4%) | 42.9 (+8%) | 44.7 (+6%) |
| 7B | SWE-Smith | Collective LB-Micro | 31.5 | 40.5 | 50.9 | 54.5 |
| 7B | SWE-Smith | ODC LB-Micro | 31.6 (+0%) | 47.5 (+17%) | 60.4 (+19%) | 60.3 (+11%) |
| 7B | SWE-Smith | ODC LB-Mini | 31.6 (+0%) | 51.2 (+26%) | 60.9 (+20%) | 60.2 (+10%) |
| 14B | LongAlign | Collective LocalSort | 20.0 | 25.0 | 29.5 | 35.5 |
| 14B | LongAlign | ODC LocalSort | 19.6 (-2%) | 25.8 (+3%) | 32.9 (+12%) | 38.5 (+9%) |
| 14B | LongAlign | Collective LB-Micro | 20.0 | 31.0 | 45.1 | 53.9 |
| 14B | LongAlign | ODC LB-Micro | 19.6 (-2%) | 31.0 (-0%) | 49.9 (+11%) | 68.9 (+28%) |
| 14B | LongAlign | ODC LB-Mini | 19.6 (-2%) | 33.4 (+8%) | 61.4 (+36%) | 69.0 (+28%) |
| 14B | SWE-Smith | Collective LocalSort | 12.3 | 14.9 | 16.5 | 18.1 |
| 14B | SWE-Smith | ODC LocalSort | 11.9 (-3%) | 15.0 (+1%) | 16.7 (+2%) | 18.2 (+1%) |
| 14B | SWE-Smith | Collective LB-Micro | 12.3 | 18.3 | 22.9 | 25.8 |
| 14B | SWE-Smith | ODC LB-Micro | 11.9 (-3%) | 20.3 (+11%) | 27.3 (+19%) | 27.5 (+7%) |
| 14B | SWE-Smith | ODC LB-Mini | 11.9 (-3%) | 23.0 (+26%) | 27.4 (+19%) | 27.6 (+7%) |
| 32B | LongAlign | Collective LocalSort | 10.3 | 12.9 | 17.3 | 20.7 |
| 32B | LongAlign | ODC LocalSort | 10.3 (+1%) | 13.6 (+5%) | 17.7 (+2%) | 21.3 (+3%) |
| 32B | LongAlign | Collective LB-Micro | 10.3 | 17.3 | 25.6 | 32.5 |
| 32B | LongAlign | ODC LB-Micro | 10.3 (+1%) | 17.0 (-2%) | 28.1 (+10%) | 39.8 (+23%) |
| 32B | LongAlign | ODC LB-Mini | 10.3 (+1%) | 18.4 (+7%) | 33.6 (+31%) | 39.4 (+21%) |
| 32B | SWE-Smith | Collective LocalSort | 7.6 | 8.2 | 9.0 | 10.5 |
| 32B | SWE-Smith | ODC LocalSort | 7.4 (-2%) | 8.3 (+1%) | 9.4 (+4%) | 11.0 (+5%) |
| 32B | SWE-Smith | Collective LB-Micro | 7.6 | 11.1 | 13.9 | 15.3 |
| 32B | SWE-Smith | ODC LB-Micro | 7.4 (-2%) | 12.6 (+13%) | 16.1 (+16%) | 16.5 (+8%) |
| 32B | SWE-Smith | ODC LB-Mini | 7.4 (-2%) | 14.3 (+29%) | 16.1 (+16%) | 16.5 (+8%) |

Table 5: Timing Data for SFT. For the percentage, ODC LB-Micro and ODC LB-Mini is comparing against Collective LB-Micro, ODC LocalSort is comparing against Collective LocalSort.

| Model | Devices | Dataset | Method | Bubble Rate (%) | | | |
|---|---|---|---|---|---|---|---|
| | | | | Minibs=1 | 2 | 4 | 8 |
| 1.5B | 8 | LongAlign | Collective LB-Micro | 66.81 | 52.63 | 35.28 | 22.08 |
| 1.5B | 8 | LongAlign | Collective LocalSort | 66.81 | 61.26 | 52.22 | 42.82 |
| 1.5B | 8 | LongAlign | ODC LB-Micro | 66.81 | 52.48 | 26.31 | 1.73 |
| 1.5B | 8 | LongAlign | ODC LB-Mini | 66.81 | 48.58 | 14.81 | 0.02 |
| 1.5B | 8 | LongAlign | ODC LocalSort | 66.81 | 58.43 | 48.29 | 39.70 |
| 1.5B | 8 | SWE-Smith | Collective LB-Micro | 52.36 | 36.28 | 20.00 | 10.74 |
| 1.5B | 8 | SWE-Smith | Collective LocalSort | 52.36 | 48.22 | 37.74 | 33.28 |
| 1.5B | 8 | SWE-Smith | ODC LB-Micro | 52.36 | 25.96 | 1.99 | 0.06 |
| 1.5B | 8 | SWE-Smith | ODC LB-Mini | 52.36 | 20.66 | 0.71 | 0.05 |
| 1.5B | 8 | SWE-Smith | ODC LocalSort | 52.36 | 46.27 | 32.40 | 29.18 |
| 7B | 8 | LongAlign | Collective LB-Micro | 63.85 | 48.31 | 30.31 | 17.77 |
| 7B | 8 | LongAlign | Collective LocalSort | 63.85 | 58.03 | 48.95 | 39.65 |
| 7B | 8 | LongAlign | ODC LB-Micro | 63.85 | 48.14 | 21.23 | 0.39 |
| 7B | 8 | LongAlign | ODC LB-Mini | 63.85 | 42.71 | 7.19 | 0.03 |
| 7B | 8 | LongAlign | ODC LocalSort | 63.85 | 54.99 | 45.11 | 36.71 |
| 7B | 8 | SWE-Smith | Collective LB-Micro | 49.70 | 33.87 | 17.41 | 9.77 |
| 7B | 8 | SWE-Smith | Collective LocalSort | 49.70 | 45.69 | 35.53 | 31.18 |
| 7B | 8 | SWE-Smith | ODC LB-Micro | 49.70 | 22.86 | 1.29 | 0.08 |
| 7B | 8 | SWE-Smith | ODC LB-Mini | 49.70 | 16.81 | 0.59 | 0.05 |
| 7B | 8 | SWE-Smith | ODC LocalSort | 49.70 | 43.75 | 30.21 | 27.08 |
| 14B | 16 | LongAlign | Collective LB-Micro | 72.28 | 57.52 | 37.75 | 24.41 |
| 14B | 16 | LongAlign | Collective LocalSort | 72.28 | 66.02 | 59.19 | 48.97 |
| 14B | 16 | LongAlign | ODC LB-Micro | 72.28 | 56.78 | 29.52 | 0.02 |
| 14B | 16 | LongAlign | ODC LB-Mini | 72.28 | 53.48 | 14.43 | 0.02 |
| 14B | 16 | LongAlign | ODC LocalSort | 72.28 | 64.20 | 52.93 | 42.46 |
| 14B | 16 | SWE-Smith | Collective LB-Micro | 56.53 | 35.00 | 18.45 | 9.30 |
| 14B | 16 | SWE-Smith | Collective LocalSort | 56.53 | 47.47 | 42.04 | 35.51 |
| 14B | 16 | SWE-Smith | ODC LB-Micro | 56.53 | 24.87 | 0.47 | 0.05 |
| 14B | 16 | SWE-Smith | ODC LB-Mini | 56.53 | 16.13 | 0.42 | 0.04 |
| 14B | 16 | SWE-Smith | ODC LocalSort | 56.53 | 45.88 | 39.27 | 32.04 |
| 32B | 32 | LongAlign | Collective LB-Micro | 73.01 | 55.19 | 35.59 | 20.43 |
| 32B | 32 | LongAlign | Collective LocalSort | 73.01 | 67.49 | 57.00 | 50.50 |
| 32B | 32 | LongAlign | ODC LB-Micro | 73.01 | 54.64 | 26.06 | 0.01 |
| 32B | 32 | LongAlign | ODC LB-Mini | 73.01 | 50.91 | 10.67 | 0.01 |
| 32B | 32 | LongAlign | ODC LocalSort | 73.01 | 64.16 | 49.55 | 37.76 |
| 32B | 32 | SWE-Smith | Collective LB-Micro | 54.03 | 32.54 | 16.69 | 8.95 |
| 32B | 32 | SWE-Smith | Collective LocalSort | 54.03 | 50.42 | 44.74 | 36.49 |
| 32B | 32 | SWE-Smith | ODC LB-Micro | 54.03 | 20.66 | 0.29 | 0.03 |
| 32B | 32 | SWE-Smith | ODC LB-Mini | 54.03 | 10.11 | 0.30 | 0.03 |
| 32B | 32 | SWE-Smith | ODC LocalSort | 54.03 | 49.25 | 40.55 | 32.16 |

Table 6: Bubble Rate Data for SFT

