# OpenReview forum: "Revisiting Parameter Server in LLM Post-Training"
_ICLR.cc/2026/Conference — ICLR 2026 Poster_

### Official Review · Reviewer_sQnr · 2025-10-27

**Soundness:** 3
**Presentation:** 3
**Contribution:** 3
**Rating:** 6
**Confidence:** 3

**Summary:**

The paper analyzes synchronization inefficiencies in Fully Sharded Data Parallel (FSDP) training when applied to LLM post-training workloads characterized by heterogeneous sequence lengths and constrained microbatch sizes. The core claim is that per-layer collective communication (all-gather in forward/backward and reduce-scatter in backward) introduces fine-grained synchronization barriers, causing straggler-induced idle bubbles. This assumption of balanced workloads holds in vision and speech models but fails in long-context supervised fine-tuning and RL workloads, where compute scales quadratically with sequence length (attention) while memory scales linearly.
The proposed method decomposes collectives into point-to-point RDMA gather and scatter-accumulate primitives, enabling asynchronous parameter fetch and gradient push at the granularity of microbatch availability. The authors demonstrate that this transformation turns FSDP into a decentralized parameter-server architecture that retains ZeRO-style memory partitioning while removing per-layer synchronization points. Synchronization is preserved only once per minibatch to maintain identical global optimizer semantics.
Evaluations on SFT (LongAlign and SWE-Smith datasets) and RL (AIME prompts with GRPO) across model scales 1.5B–32B show up to 36% throughput improvement and strong correlation with predicted imbalance (bubble rate).

**Strengths:**

- Precise identification of bottleneck in sharded data parallelism. The per-layer max-runtime aggregation inherently propagates worst-case slowdowns, validated with measurements showing up to 50% idle bubbles.
- The introduced method preserves DP correctness constraints: a) parameter, gradient, optimizer sharding identical to FSDP. b) deterministic synchronous update at minibatch boundaries, and c) No model semantic change.
- Lightweight integration: <100 LOC modification using CUDA-IPC + NVSHMEM RDMA via Triton-Distributed.

**Weaknesses:**

-  Benchmarking shows significant bandwidth degradation vs NCCL rings when D > G.
Inter-node performance constraints remain unresolved in this work.
- The degree of compute–communication overlap is asserted but not rigorously quantified (e.g., timeline traces, GPU utilization per stage). This limited control-flow characterization does not give much confidence to the reader on the effectiveness in overlapping.
- Comparison set omits strong baselines. There is no experimentation vs ZeRO++ hierarchical collectives, scheduling-based overlap strategies, or pipeline-staleness variants.
- RL system constraints to some extend weaken conclusions. To be specific, forcing equal samples per device, reduces the effectiveness of proposed minibatch-level packing (LB-Mini).
- Claims of benefits growing with scale lack large-cluster support are not supported by the results since the scaling experiments are not extended to >32 GPUs

**Questions:**

- What is the asymptotic slowdown of ODC communication when inter-node degree grows? Can you provide a model predicting crossover vs collectives?
- Does the minibatch-level synchronization impose idle bubbles in the best-case balanced regime? If so, is dynamic minibatch staggering feasible?
- In RL, how would delayed optimizer updates or local advantage normalization interact with async shard availability?
- What is the concurrency behavior of the gradient accumulation daemon at high client cardinality? Any queue saturation or tail latency concerns?

---

> ### Author Response · Authors · 2025-11-17
>
> > Challenges on inter-node performance of ODC primitives
>
> We agree that this is a limitation of our current solution, and we will work on optimizing this in the future. Moreover, as discussed in Section 6.1 (Line 449) and empirically demonstrated in Appendix E (Figure 12), this limitation can be effectively mitigated by using a ZeRO++ style hybrid sharding strategy. Our results in Appendix E show that ODC still achieves significant performance improvements (up to 28%) over collectives in this hybrid-sharding setup, demonstrating its effectiveness.
>
> > The degree of compute–communication overlap is asserted but not rigorously quantified.
> > What is the asymptotic slowdown of ODC communication when inter-node degree grows? Can you provide a model predicting crossover vs collectives?
>
> These are insightful questions that cut to the core of the limitation on cross-node communication. We can address them with quantitative models:
>
> **Asymptotic Performance Comparison**:
> When inter-node communication becomes the bottleneck, the total traffic of ODC is approximately proportional to $D K$, whereas the ring-collective’s traffic is roughly $D K/G$, where D is the total number of devices and G is the number of GPUs per node.
> Assuming G=8, ODC therefore incurs about 8× higher inter-node traffic.
>
> **Impact on End-to-End Runtime with Overlap**:
> In practice, overlapping communication and computation substantially mitigates this effect.
> Assuming microbatch runtime $T = \max(T_{\text{comm}}, T_{\text{comp}})$, where
> $T_{\text{comm}} \approx 6M / B_{\text{comm}}$​ (2 all-gathers and 1 reduce-scatter) and $T_{\text{comp}} \ge 8sM / B_{\text{comp}}$, with s, M, and B denoting the number of tokens, parameters, and bandwidth, respectively. The crossover point between compute-bound and communication-bound regimes satisfies $s \le 6B_{\text{comp}} / 8B_{\text{comm}}$
>
> Empirically, in our settings, the crossover microbatch size is approximately 1k tokens for collectives and 8k tokens for ODC. Thus, for microbatches larger than 8k tokens (common in post-training settings with sequence packing), ODC’s communication slowdown is effectively hidden. For settings with smaller number of microbatches, we recommend a ZeRO++-style hybrid sharding configuration to constrain ODC communication within each node.
>
> The following figure shows NSight profiling forward computation overlapped with ODC Gather primitives with s=16k and 16 GPUs, which matches our expectation quite well.
>
>
> [Communication Overlapping Profile](https://i.imgur.com/uGOb5MI.png)
>
>
> > Comparison set omits strong baselines. There is no experimentation vs ZeRO++ hierarchical collectives, scheduling-based overlap strategies, or pipeline-staleness variants.
>
>
> Thank you for this suggestion. We did evaluate ZeRO++ because it fits ODC well when microbatch size is low (because in ZeRO++ parameters and gradients are sharded intra-node, avoiding slow inter-node ODC communications) in Appendix E (Figure 12). Our results show ODC provides significant speedup in this configuration.
>
>
> Can you elaborate a bit on the other baselines you mentioned like scheduling-based strategies and pipeline-staleness?
>
>
> > Does the minibatch-level synchronization impose idle bubbles in the best-case balanced regime? If so, is dynamic minibatch staggering feasible?
>
> If the 'best-case' indicates cases where the workload is fully balanced, the minibatch-level synchronization does not introduce idle bubbles. Synchronization merely ensures that the reduce-accumulate ODC primitive has completed across all devices. it incurs negligible overhead and effectively acts as a no-op.
>
> > In RL, how would delayed optimizer updates or local advantage normalization interact with async shard availability?
>
>
> We appreciate this question and would like to clarify our understanding. If the question refers to the multiple PPO steps in an iteration and the normalization in GRPO, these do not interfere with ODC:
> - Each PPO step corresponds to a minibatch, which in turn contains multiple microbatches—thus behaving analogously to SFT training.
> - Advantage computation occurs before any forward/backward computation, and therefore remains unaffected by the asynchronous shard availability.
>
> We would be happy to provide additional clarifications if more specific details about the RL setup are provided.
>
> > What is the concurrency behavior of the gradient accumulation daemon at high client cardinality? Any queue saturation or tail latency concerns?
>
> This is an excellent question. In our design, multiple clients can concurrently push gradients to the same server, and each client can simultaneously push to multiple servers to maximize bandwidth utilization. To prevent server-side congestion, we limit the number of concurrent pushes from each client, which implicitly bounds the number of simultaneous RDMA receive operations on the servers. Empirically, we have observed stable execution times for scatter–accumulate operations.

---

> > ### Author Response · Authors · 2025-11-28
> >
> > As the discussion period is nearing its close, we would greatly appreciate it if you could take a brief moment to review our responses and confirm whether they satisfactorily resolve your questions. If our clarifications have improved your confidence in the paper, we would be sincerely grateful if you could consider updating your score accordingly.

---

> > > ### Author Response · Authors · 2025-12-02
> > >
> > > We apologize for previously overlooking the last two stated weaknesses. We provide further clarification below:
> > >
> > > > Regarding the concern that RL system constraints weaken the conclusions:
> > >
> > > The requirement of equal samples per device is a limitation specific to the current implementation of VeRL. This constraint is not inherent to RL systems in general and could be relaxed with moderate engineering effort. We did not pursue such modifications because they fall outside the scope of our work. Importantly, other RL frameworks do not necessarily impose the same restriction.
> > >
> > > > Regarding the limited experiment scale (32 GPUs):
> > >
> > > We agree that larger-scale experiments would further strengthen the empirical evidence. Unfortunately, such resources are not accessible to most researchers, including us. From a theoretical perspective, increasing the number of devices would amplify the imbalance effect, because end-to-end latency is governed by the slowest device. Consequently, ODC would be even more beneficial at larger scales. This trend is illustrated in Figure 10 (Acceleration Ratio vs. Number of Devices).

---

### Official Review · Reviewer_7zfy · 2025-10-28

**Soundness:** 3
**Presentation:** 3
**Contribution:** 2
**Rating:** 2
**Confidence:** 5

**Summary:**

The paper proposed on-demand-communication, which separate worker and parameter sever as old-fashion PS.

**Strengths:**

1. good idea to call back to parameter server. Personally I like this call-back.
2. clear presentation and experimental results

**Weaknesses:**

1. With PS, one major issue is we have inconsistency of parameters (i.e. parameter delay). For example, two GPUs , GPU0 pull the weights before some xyz updates, then GPU1 pull the same weights with xyz updates, then they are training on different weights but same iteration number, This parameter inconsistency will hurt model convergences. There is no discussion on model convergence in the whole paper.

2. Indeed, FSDP/ZeRO they already incorporated async parameter pre-fetch, hiding weights updates behind OS (optimizer states) updates etc. This communication hiding mechanism works pretty well and I don't see the real motivation for deleting all the global collective into pair-wise p2p communication. The statement of this paper in L134 "but this overlap does not remove the underlying synchronization points." is wrong, because, every GPU can pre-fetch a number of layers weights all together at once, they don't need to wait for each other at every layer's fwd or bwd

3. In real world post-training, we do either batching or padding to make every GPU has same sequence length of input, thus the straggler problem this paper try to solve does not exists in real-world application scenarios.

**Questions:**

Try to profile some real world workload, and real-world system setting (e.g., how to do data loading, data preprocessing (padding, batching), distributed FSDP+prefetch, LoRA, etc), for post training is a good way to find where is the real bottleneck.

---

> ### Author Response · Authors · 2025-11-17
>
> We sincerely thank the reviewer for the critical review of our work on ODC. We appreciate this opportunity to clarify our method's design, its motivation, and the specific problem it addresses.
>
> ## W1: Inconsistency of parameters.
>
> We would like to respectfully point out a central point: **ODC preserves the exact synchronous training semantics of FSDP**. There is no parameter inconsistency or staleness. This is because ODC retains the synchronous optimizer step at the end of each minibatch. As illustrated in Figure 1, all forward and backward passes for all microbatches (within a single gradient accumulation phase) must complete before the `optimizer.step()` is called. Therefore, all microbatches within the same global iteration compute gradients based on the exact same set of parameters.
>
> We empirically validated this in Figure 14, which shows identical training loss curves between ODC and the FSDP baseline, confirming the unchanged training semantics.
>
> The confusion may arise from the term of "Parameter Server." We use this term to describe the communication pattern (i.e., fetch-on-demand) and its resulting robustness to workload imbalance, not to imply asynchronous updates. Actually, in many previous work like BytePS (OSDI '20), PS can support both synchronous and asynchronous training semantics.
>
>
> ## W3: we do either batching or padding to make every GPU has same sequence length of input… The problem does not exist and authors should find where is the real bottleneck.
>
>
> We feel there is a confusion on padding. The key point is, **padding does not solve imbalance**. Padding adds unused tokens to shorter sequences, but **the computation on these padded tokens is wasted**. Instead, ODC fills the bubble with useful computation from next microbatch. In other words, though padding can align the sequence length, the run time is still the same as unpadded imbalanced situations.
>
>
> **Packing (also known as dynamic batching) is indeed effective at reducing load imbalance. However, the imbalance problem fundamentally remains,** because it is impossible to align both memory $O(s)$ and the compute $O(s^2)$ in packing. Even if multiple microbatches have the same number of tokens, their compute time can significantly differ. For example, in a microbatch that contains a single sequence with the maximum sequence length, no combination of shorter sequences can match its runtime. Moreover, for long-tailed sequence-length distributions, which is common in many post-training workloads, it is often impossible to pack enough short sequences to match the compute cost of the few very long sequences.
>
>
> Importantly, **the packing algorithm used in our experiments is already highly optimized**. As shown in Figure 9, **our baseline (Collective LB-Micro) is substantially faster than VeRL’s current load-balancing implementation** (Collective LB-Native). Yet even with this strong baseline, Tables 4 and 6 show that the bubble rate can still reach up to 50% for a minibatch of four sequences, demonstrating that significant imbalance persists despite packing.
>
> **Workload imbalancing is a non-trivial bottleneck of LLM post training**, supported by **various studies on top venues like LongAlign (ACL’ 24), WLB-LLM (OSDI’ 25), etc**. And ODC is an approach from a different angle - we relax synchronizations to a more coarse granularity.
>
> ## W2: async parameter pre-fetch works pretty well
>
> We thank the reviewer for this important point. While asynchronous prefetching is an essential optimization for hiding communication latency, we wish to clarify that **it does not solve the fundamental problem of workload imbalance**, which stems from synchronization barriers:
>
> - **The Microbatch Barrier Persists**: Prefetching overlaps communication for subsequent layers, but it cannot eliminate the initial synchronization point at the start of every microbatch. All devices must begin the forward pass of a microbatch together. This initial all-gather for the first layer acts as a hard barrier, forcing faster devices to wait for stragglers before any computation in the microbatch can begin.
>
> [Example of Prefetch](https://i.imgur.com/exeg3ku.png)
>
> - **Synchronization is The Major Problem**: Because all devices are synchronized at the start of each micro-batch, any time difference in completing the previous micro-batch translates directly into idle time ("bubbles"). Prefetching can hide the cost of the all-gather after this point, but it cannot hide the initial waiting time caused by the workload imbalance itself.
>
> - **Our Experiments Already Include Prefetching: Crucially, all of our baseline experiments were conducted with FSDP's asynchronous prefetching enabled.** The significant speedups we demonstrate with ODC are therefore achieved on top of this optimization. This empirically proves that prefetching is insufficient to resolve the inefficiency caused by workload imbalance.
>
> We are happy to provide any further clarification.

---

> ### Author Response · Authors · 2025-11-28
>
> Thank you again for your critical review. We carefully addressed the points of confusion raised during the discussion phase, and we would sincerely appreciate it if you could revisit our responses and update your review should they help clarify your concerns.
>
> To further substantiate our claims, **we conducted two additional experiments**, with one presented below.
>
> ## Addressing W3: Padding to the Same Length Does Not Improve Performance
>
> In our original experiments, **packing-based load balancing was already applied to the baseline (see Section 5.2)**. In this follow-up experiment, we introduce an additional step: for each microbatch across devices, **we pad the number of tokens so that every device reaches the same token count** as the device with the maximum number of useful tokens in that microbatch. The results are shown below.
>
> | Method                                          | Minibatch Size = 1 | 2            | 4             | 8             |
> | ----------------------------------------------- | ------------------ | ------------ | ------------- | ------------- |
> | Packing (baseline LB-Micro in main experiments) | 0.052 |0.073 |0.102 |0.119 |
> | Packing + Pad to Same Length                    | 0.052 (+0%)|0.074 (+1%) |0.096(-6%) |0.110 (-7%) |
> | Packing + ODC                                   | 0.052 (+0%)|0.081 (+10%) |0.135 (+32%)|0.137 (+15%) |
>
> *Training throughput (samples per GPU per second) for the 7B model on LongAlign.*
>
> As shown, **padding** introduces additional non-informative tokens and therefore does not outperform the baseline packing strategy. Its performance remains comparable to packing-only methods and, in several cases, is worse. This is because padding merely creates an artificial appearance of workload balance by filling idle bubbles with redundant computation, whereas our proposed method replaces these bubbles with useful computation.
>
> While **packing** (also referred to as dynamic batching) remains a valuable and widely adopted technique, it still leaves substantial room for improvement. For this reason, we treat packing as a strong baseline and focus our contributions on demonstrating more effective approaches to mitigating workload imbalance.
>
> ## Addressing W2: Prefetch does not remove the microbatch-level communication barrier
>
> We compare vanilla FSDP with collective communication, which prefetches one layer by default, against configurations that prefetch a variable number of layers $n$. This is implemented by wrapping $n$ consecutive layers into a single parameter group in FSDP. The results are summarized in the following table.
>
> | Method       | Prefetch Layers                                   | Troughput (samples/GPU/s) | Memory (GB)   |
> | ------------------|----------------------------- | ------------------ | ------------ |
> | FSDP+Collective |1 (default)| 0.105|37.7|
> | FSDP+Collective|2 |0.106 (+0%)|38.6 (+2%)|
> | FSDP+Collective| 4|0.104 (-1%)|40.4 (+7%)|
> | FSDP+Collective| 8|0.105 (+0%)|42.6 (+13%)|
> | FSDP+Collective| 14 (50% of model)| 0.094 (-11%)|49.4 (+31%)|
> | FSDP+ODC |1 (default)|0.135 (28%) |36.32|
>
>
> *7B model on LongAlign Dataset.*
>
> As shown, increasing the number of prefetched layers has negligible impact on throughput. This confirms our earlier claim that the **microbatch-level synchronization barrier persists due to the mandatory synchronization before the first layer**, which cannot be hidden by prefetching additional layers.
>
> We need to note that prefetching sacrifices memory. In extreme cases when we prefetch every layer, FSDP falls back to ZeRO Stage 1. Moreover, prefetching too many layers actually has negative throughput impact, as the initial fetch of the first parameter group, which can not be overlapped by compute, takes too long.

---

> > ### Author Response · Authors · 2025-11-28
> >
> > As the discussion period is nearing its close, we would greatly appreciate it if you could take a brief moment to review our responses and confirm whether they satisfactorily resolve your questions. If our clarifications have improved your confidence in the paper, we would be sincerely grateful if you could consider updating your score accordingly.

---

### Official Review · Reviewer_ycWJ · 2025-10-30

**Soundness:** 4
**Presentation:** 4
**Contribution:** 4
**Rating:** 8
**Confidence:** 3

**Summary:**

This paper addresses a critical limitation of modern data-parallel training (e.g., FSDP) in LLM post-training, where variable sequence lengths cause severe workload imbalance and render collective communication inefficient due to strict synchronization barriers. The authors propose ODC, a novel adaptation of the PS paradigm within FSDP that replaces collective all-gather/reduce-scatter with point-to-point communication. By decoupling device synchronization to the minibatch level and enabling asynchronous gradient exchange, ODC eliminates workload-induced stalls and facilitates fine-grained load balancing. Experiments across diverse LLM post-training tasks show consistent gains in device utilization and throughput, demonstrating that PS-style communication is not obsolete, but better suited to real-world LLM training dynamics. The approach is practical and novel.

**Strengths:**

- Novelty: The proposed method adapts the parameter server concept into the FSDP framework by replacing collective all-gather/reduce-scatter operations with point-to-point communication, thereby reducing synchronization overhead.
- Practicality: The authors implemented ODC and provided extensive experimental results demonstrating its reliability.
- Readability: The paper is well-structured and easy to follow.

**Weaknesses:**

No obvious drawbacks.

**Questions:**

- Under the setting of activation recomputation, is ODC still applicable?
- What are the potential limitations of ODC compared to FSDP?

---

> ### Author Response · Authors · 2025-11-17
>
> > Under the setting of activation recomputation, is ODC still applicable?
>
> Yes, ODC remains fully applicable under activation recomputation. Conceptually, with activation recomputation, the backward pass of a layer can be viewed as a forward pass involving parameter gathering, followed by a backward pass that scatters gradients. This interpretation aligns naturally with the ODC communication model.
> In fact, all our SFT and RL experiments are conducted with gradient checkpointing (activation recomputation) to manage memory consumption in long-context training scenarios, demonstrating ODC’s compatibility with this setting in practice.
>
> > What are the potential limitations of ODC compared to FSDP?
>
> The primary limitation of ODC, as discussed in Section 6, lies in the current level of system optimization. Existing implementations of cross-node ODC primitives are not yet as highly optimized as expert-tuned collective libraries such as NCCL. However, with dedicated optimization efforts similar to those applied to collective primitives, we believe ODC can achieve better efficiency in future work.

---

### Official Review · Reviewer_xMDa · 2025-10-31

**Soundness:** 3
**Presentation:** 3
**Contribution:** 3
**Rating:** 6
**Confidence:** 3

**Summary:**

This paper proposes On-Demand Communication (ODC), which removes per-layer synchronization barrier from training LLM in FSDP. Because of the imbalanced workload in post-training, with a simplified load balancing strategy, ODC helps speedup over FSDP by replacing collective calls with P2P communication.

**Strengths:**

- Well-motivated and straightforward to implement.
- An excellent complement to FSDP that only requires changing communication operators—theoretically and experimentally equivalent (minor precision differences from batch size variations have minimal impact).

**Weaknesses:**

The practical applications of ODC may be quite limited, requiring specific scenarios with load imbalance. In SFT, the paper only tested LongAlign and SWE-Smith; in RL, updating the actor is not the main bottleneck, and currently, partial-rollout or fully asynchronous training are more commonly used to improve overall system throughput.

**Questions:**

Can ODC extend to support MoE models in FSDP? Though not efficient as expert parallel method, it might be better than the standard FSDP?

---

> ### Author Response · Authors · 2025-11-17
>
> We appreciate the reviewer’s thoughtful comments. Here are our responses:
>
> > The practical applications of ODC may be quite limited, requiring specific scenarios with load imbalance. In SFT, the paper only tested LongAlign and SWE-Smith; in RL, updating the actor is not the main bottleneck, and currently, partial-rollout or fully asynchronous training are more commonly used to improve overall system throughput.
>
> It is correct that the acceleration from ODC depends on the degree of workload imbalance. However, we argue that such imbalance is a pervasive phenomenon in post-training of LLMs, arising naturally from variable sequence lengths. Therefore, ODC addresses a broad and practically relevant challenge rather than a niche case.
>
> Regarding SFT, we believe the workload imbalance generally exists and would be more critical especially when the context length is long. For example, most LLM agents require super long context windows to execute complex real-world tasks, and often use SFT as a cold start. As LLM agent is one of the hottest topics in both research and products, we believe our work can provide a better solution in such scenarios.
>
> Regarding RL fine-tuning, the most straightforward benefit is on the training side, namely faster updating actor/critic and faster computing old_log_prob/ref_log_prob. The training time takes about 30~60% of the total time. Although many works are trying to acclerating the rollout, we argue that training side also matters a lot. Most importantly, with ODC, we can potentially resolve the long-tail issues for rollout. Specifically, when one device completes its rollout, it can directly start the training without waiting other devices to join! In this sense, we believe ODC can open a lot of potentials as we discussed in the Future Work (Section 6.2).
>
> As ODC is a lightweight and general communication primitive that can be seamlessly integrated into existing frameworks, we believe it will benefit many scenarios due to its PS nature and FSDP's memory efficiency.
>
> > Can ODC extend to support MoE models in FSDP? Though not efficient as expert parallel method, it might be better than the standard FSDP?
>
>
>
> This is an excellent point. The applicability of ODC to MoE models depends on how the model is wrapped within FSDP. If  the wrapping is performed at the layer level for relatively small models, FSDP (and therefore ODC) behaves identically to the dense case. However, when the wrapping occurs at the expert level (which is possible as one layer in MoE model can be too big), ODC becomes more advantageous, since MoE models involve more frequent synchronization than dense models. By eliminating these synchronization barriers, ODC can provide additional efficiency gains beyond what standard FSDP achieves.

---

> > ### Author Response · Authors · 2025-11-28
> >
> > Thanks again for the support on our paper. As the discussion period is nearing its close, we would greatly appreciate it if you could take a brief moment to review our responses and confirm whether they satisfactorily resolve your questions. We are happy to provide more clarifications if necessary.

---

> > > ### Comment · Reviewer_xMDa · 2025-11-28
> > >
> > > Thanks for your response. ODC does seem promising in certain scenario. But my major concern is that ODC is limited in the framework of FSDP, while a clear trend is that the model parallel and MoE models are taking over. It’s also not very reasonable to expect the authors to directly compare against Megatron.
> > >
> > > Though I will remain my score, I think ODC makes a good complement to FSDP.

---

> ### Author Response · Authors · 2025-12-02
>
> We sincerely appreciate your support for ODC. We believe that FSDP/ZeRO-3 remains a strong fit for medium-scale training due to its simplicity and the fact that it requires no modifications to model code and adapts to all models despite model architecture. Consequently, FSDP continues to be widely adopted within the research community.
>
> That said, we agree that hybrid parallelism strategies such as DP/PP/TP/EP (i.e., 4D parallelism) are more suitable for ultra-large-scale training workloads. Perhaps, this may simply be a matter of scale: 4D parallelism is better aligned with the demands of very large-scale training, whereas FSDP provides a practical and effective solution at medium scale. We believe that challenges at both scales are important and valuable for the systems research community.

---

### Author Response · Authors · 2025-11-17

We sincerely thank all reviewers for their feedback on our paper. Please find our responses to individual comments below.

---

### Meta-Review · Area_Chair_U5ts · 2025-12-31

**Summary:**

This paper revisits the parameter server  paradigm in the context of large language model post-training and proposes On-Demand Communication as an adaptation of PS-style point-to-point communication within the FSDP/ZeRO-3 framework. The key insight is that workload imbalance induced by variable sequence lengths undermines the efficiency of per-layer collective communication, and that relaxing synchronization from the layer level to the minibatch level can significantly improve device utilization. Reviewers broadly agree that the paper identifies an important and timely systems bottleneck and presents a technically sound and well-implemented solution. Extensive experiments across supervised fine-tuning and reinforcement learning settings demonstrate consistent throughput gains, in some cases up to 36%. While opinions differ regarding the breadth of applicability and long-term positioning of the approach, the paper is generally viewed as a meaningful systems contribution.

**Reviewer Concerns:**

The reviewers' primary concerns centered on the following aspects:

(1) The scope and generalizability of ODC, particularly its tight coupling with fully sharded data parallelism  and its relevance in light of emerging training paradigms such as hybrid parallelism and Mixture-of-Experts.

(2) Cross-node communication efficiency and the absence of validation at large scale (e.g., beyond 32 GPUs).

(3) The comprehensiveness and competitiveness of baseline comparisons, including hierarchical collective operations and alternative strategies for overlapping or scheduling communication and computation.

(4) Ambiguities regarding training semantics and convergence behavior compared to traditional parameter server architectures.

**Reviewer Scores:**

The reviewer scores reflect a mixed but overall positive assessment. Reviewer ycWJ (score 8) provided a strong accept recommendation, highlighting the paper’s clear identification of synchronization bottlenecks in FSDP-based LLM post-training and the practical effectiveness of the proposed ODC approach. Reviewers xMDa and sQnr (score 6) rated the paper marginally above the acceptance threshold, acknowledging the technical soundness and empirical throughput gains while expressing reservations regarding the scope of applicability, baseline coverage, and scalability to larger or hybrid-parallel training settings. In contrast, Reviewer 7zfy (score 2) remained strongly negative, raising concerns about the motivation and parameter-server semantics that were not fully reconciled despite extensive rebuttal. Overall, the majority of reviewers are supportive, with one dissenting opinion, and the scores show convergence toward acceptance as a systems-oriented contribution.

---

### Decision · Program_Chairs · 2026-01-26

Accept (Poster)